# Unsupervised Hierarchical Clustering of Head and Neck Cancer Patients by Pre-Treatment Plasma Metabolomics Creates Prognostic Metabolic Subtypes

**DOI:** 10.3390/cancers15123184

**Published:** 2023-06-14

**Authors:** Ronald C. Eldridge, Zhaohui S. Qin, Nabil F. Saba, Madelyn C. Houser, D. Neil Hayes, Andrew H. Miller, Deborah W. Bruner, Dean P. Jones, Canhua Xiao

**Affiliations:** 1Nell Hodgson Woodruff School of Nursing, Emory University, Atlanta, GA 30322, USA; madelyn.elizabeth.crawford@emory.edu (M.C.H.); deborah.w.bruner@emory.edu (D.W.B.); cxiao2@emory.edu (C.X.); 2Department of Biostatistics and Bioinformatics, Rollins School of Public Health, Emory University, Atlanta, GA 30322, USA; zhaohui.qin@emory.edu; 3Winship Cancer Institute, Emory University, Atlanta, GA 30322, USA; nfsaba@emory.edu; 4Department of Medicine, UT/West Institute for Cancer Research, University of Tennessee Health Science Center, Memphis, TN 38163, USA; neil.hayes@uthsc.edu; 5Department of Psychiatry and Behavioral Sciences, School of Medicine, Emory University, Atlanta, GA 30322, USA; amill02@emory.edu; 6Division of Pulmonary, Allergy and Critical Care Medicine, Emory University, Atlanta, GA 30322, USA; dpjones@emory.edu

**Keywords:** metabolomics, head and neck cancer, biomarkers, survival, clustering, fatty acids, smoking, human papillomavirus, arginine and proline, galactose

## Abstract

**Simple Summary:**

There is a need to identify new translational prognostic biomarkers in head and neck cancer. Metabolomics, the study of small molecules resulting from cellular metabolism, is an emerging and promising field regarding head and neck cancer. We performed metabolomics on patients’ blood prior to treatment and found that it can divide patients into high-risk and low-risk groups based on their cancer progression and survival. We believe our study provides compelling results to consider metabolomics as a translational prognostic biomarker and that it may offer novel information for patient risk stratification. With continued research, we hope to gain a fuller understanding of how metabolomics may aid in the early detection, prognosis, treatment monitoring, and targeted therapies of head and neck cancer.

**Abstract:**

There is growing evidence that the metabolism is deeply intertwined with head and neck squamous cell carcinoma (HNSCC) progression and survival but little is known about circulating metabolite patterns and their clinical potential. We performed unsupervised hierarchical clustering of 209 HNSCC patients via pre-treatment plasma metabolomics to identify metabolic subtypes. We annotated the subtypes via pathway enrichment analysis and investigated their association with overall and progression-free survival. We stratified the survival analyses by smoking history. High-resolution metabolomics extracted 186 laboratory-confirmed metabolites. The optimal model created two patient clusters, of subtypes A and B, corresponding to 41% and 59% of the study population, respectively. Fatty acid biosynthesis, acetyl-CoA transport, arginine and proline, as well as the galactose metabolism pathways differentiated the subtypes. Relative to subtype B, subtype A patients experienced significantly worse overall and progression-free survival but only among ever-smokers. The estimated three-year overall survival was 61% for subtype A and 86% for subtype B; log-rank *p* = 0.001. The association with survival was independent of HPV status and other HNSCC risk factors (adjusted hazard ratio = 3.58, 95% CI: 1.46, 8.78). Our findings suggest that a non-invasive metabolomic biomarker would add crucial information to clinical risk stratification and raise translational research questions about testing such a biomarker in clinical trials.

## 1. Introduction

Recent treatment advances for head and neck squamous cell carcinoma (HNSCC) have improved patients’ survival but at a large cost to their quality of life [1]. The recommended regimen of chemoradiotherapy causes up to 80% of HNSCC patients to experience debilitating side effects such as mucositis, dysphagia, malnutrition, speech problems, lymphedema, and cognitive decline [2,3,4,5,6,7]. Additionally, more than 50% of these patients remain afflicted years after treatment [8]. This suffering has motivated prognostic biomarker research to help expedite clinical trials of less toxic targeted therapies (e.g., epidermal growth factor receptor inhibitors) and de-escalated standard therapies (e.g., lower doses of radiation) [9].

Human papillomavirus (HPV), and to a lesser extent, tobacco use, are the most widely translated prognostic markers in HNSCC. In 2010, Ang et al. proved that HPV status, when combined with smoking pack-years, could stratify oropharyngeal SCC patients into low, intermediate, and high-risk groups with three-year survivals of 93%, 71%, and 46%, respectively [10]. Their findings spawned a decade of de-escalated therapeutic research to determine if patients deemed low-risk—those who are HPV-positive with a minimal history of tobacco use—could be spared the long-term toxicity of standard treatment. Despite recent setbacks [11,12], this remains an active area of research [13,14]. More recently, circulating HPV DNA has garnered interest as a non-invasive biomarker of post-treatment surveillance, since it can accurately predict tumor recurrence [15]. However, its clinical utility would be limited to HPV-positive oropharyngeal SCC—approximately 30% of the total HNSCC patient population [16].

Other HNSCC prognostic markers being studied include genetic mutations, molecular variations, immunologic signaling, and tumor imaging, but while this is promising, none have been proven to optimize therapy. Functional loss of p53, a tumor suppressor gene, is associated with poor prognosis [17]; however, its high frequency of mutation—upwards of 85% in HPV-unrelated HNSCC [18]—raises questions about its discriminating specificity. Overexpression of the epidermal growth factor receptor (EGFR), a well-studied tyrosine kinase receptor, has shown a prognostic ability, however, similar to the loss of p53, it is also quite common (~90%) in HNSCC [19]. Furthermore, there is no standardized antibody or staining protocol to measure EGFR, and overexpression of EGFR is not consistently associated with a response to EGFR-targeted therapies (e.g., cetuximab) [20], raising additional concerns about its future clinical use. The immunosuppressive nature of the HNSCC microenvironment, relative to cancers at other tissue sites, makes immunologic biomarkers interesting targets. Programmed cell death-1 (PD-1) and its primary ligand (PD-L1) are transmembrane signaling receptors that negatively regulate the adaptive immune response (e.g., cytotoxic T cells). PD-1 and PD-L1 are checkpoint inhibitors being studied as targeted HNSCC immunotherapy after the failure of standard therapy [21,22]. Upregulated PD-L1 is associated with a worse prognosis but more evidence is needed to predict which patients will respond to anti-PD-1 monoclonal antibody immunotherapy [23]. Non-specific immunosuppressive cytokines and chemokines (e.g., IL-10 and TGF-beta) show some ability as prognostic biomarkers of tumor-mediated immunomodulatory signaling but results remain inconsistent [24]. Lastly, cross-sectional tumor imaging via magnetic resonance imaging (MRI), computed tomography (CT), or positron emission tomography/computed tomography (PET/CT), plays an important role in the diagnosis and clinical management of HNSCC. The imaging of HNSCC often provides crucial information for TNM staging [25]. Outside of staging, functional imaging by 18-fluorodexyglucose PET/CT provides a highly sensitive measure of tumor metabolic activity via glucose uptake. Patients whose tumors show higher levels of metabolic activity tend to have a worse prognosis [26], but questions remain about quantifiability and low specificity since non-cancerous tissue and immune cells in the microenvironment also have a high glucose uptake.

Metabolic biomarkers have, in general, been overlooked in HNSCC translational research compared to their genomic, molecular, immunologic, and imaging counterparts. There is, however, growing evidence that the metabolism plays an outsized role in activating the host’s immune response [27] and that it is deeply intertwined with HNSCC progression and survival [28]. In 2020, five independent studies found differential expression of metabolic genes to be prognostic of patient survival [29,30,31,32,33]. While promising, translating a gene-based metabolic biomarker poses some challenges. First, the within-tumor metabolic heterogeneity of HNSCC is extreme, [34] amplifying the probability of sampling bias; moreover, direct sampling of the tumor is costly, invasive, and burdensome on the patient [35,36]. Second, since most metabolic regulation occurs downstream of the genome, [37] a gene-based biomarker is prone to miss vital metabolic transitions and patterns. A non-invasive alternative is to measure circulating metabolites in patients’ blood. Recent advances in high-resolution metabolomics allow for comprehensive quantification of the host’s metabolism and systemic metabolic signaling. However, as an emerging field, little is known about circulating metabolite patterns in HNSCC patients and their clinical potential. In the following study, we used multivariate unsupervised hierarchical clustering of plasma metabolomics to discover metabolic subtypes from a heterogeneous group of 209 HNSCC patients. We sought to answer three questions: (1) Are there circulating metabolic differences in HNSCC patients? Secondly, (2) if so, are those metabolic differences prognostic of outcomes, namely survival? Lastly, (3) what are the implications for a potential non-invasive metabolomic biomarker in HNSCC?

## 2. Materials and Methods

### 2.1. Study Cohort

This was a prospective cohort of 209 HNSCC patients recruited from 2013 to 2016 at radiation oncology clinics affiliated with Emory University Hospital before undergoing intensity-modulated radiation therapy (IMRT) or IMRT with concurrent platinum-based chemotherapy [38]. The inclusion criteria were histological HNSCC with no distant metastasis, ≥21 years of age, and no evidence of uncontrolled metabolic, hematologic, cardiovascular, renal, hepatic, or neurologic disease. Patients with simultaneous primaries or major psychiatric disorders were excluded. Demographic and clinical variables were collected through chart review and standardized questionnaires at study entry. Prior to IMRT, blood was collected into chilled EDTA tubes for the immediate isolation of plasma. Plasma was separated by centrifugation at 1000× *g* for 10 min at 4 °C, then aliquoted into siliconized polypropylene tubes and stored at −80 °C.

### 2.2. Overall and Progression-Free Survival

Vital status and disease progression were ascertained by linking our patient cohort to the Georgia Comprehensive Cancer Registry maintained at Emory University. Overall survival was defined as the time from study entry to death by any cause. Progression-free survival was defined as the time from study entry to HNSCC progression noted in the medical chart or death. Patients who did not experience the outcome were censored at the last date of contact up to June 2022. We were unable to follow 20 subjects and they were excluded from the survival analyses.

### 2.3. High-Resolution Untargeted Metabolomics (HRM) of Blood Plasma

Our HRM approach used established liquid chromatography-mass spectrometry (LC-MS) protocols developed at the Emory Clinical Biomarkers Laboratory [39,40] (Appendix A). Frozen aliquots were thawed and extracted with ice-cold acetonitrile. Supernatants were added to a 4 °C autosampler in random order. Each sample was divided and analyzed in triplicate (10 μL) using dual chromatography separation: hydrophilic interaction liquid chromatography (HILIC) with positive electrospray ionization and 18-carbon (C18) with negative ionization. Analyte separation was achieved using a 2.1 mm × 100 mm × 2.6 μm Accucore column (Thermo Scientific, Norcross, GA, USA) and a gradient elution of 2% of formic acid, water, and acetonitrile starting at 10%, 10%, and 80%, for 1.5 min with a linear increase to 10%, 80%, and 10% at 6 min and held for 4 min per injection.

The high-resolution Fourier transform Orbitrap mass spectrometer (Dionex Ultimate 3000, Q-Exactive HF, Thermo Scientific) operated in full scan mode at 120,000 resolution and a mass-to-charge ratio (m/z) range of 85.0000–1275.0000. Probe temperature, capillary temperature, sweep gas, and S-Lens RF levels were maintained at 200 °C, 300 °C, 1 arbitrary unit (AU), and 45 AU, respectively. Positive tune settings for sheath gas, auxiliary gas, sweep gas, and spray voltage settings were 45 AU, 25 AU, and 3.5 kV, respectively. Raw data were extracted and aligned using apLCMS [41] and xMSanalyzer [42] separately for the HILIC-positive and C18-negative chromatography runs. Mass, retention time, and ion intensity were measured for each uniquely identified m/z spectral feature. The m/z features with a median coefficient of variation (CV) within technical replicates ≥75% were removed. The remaining feature intensities were median-summarized across sample triplicates with the requirement that ≤1 replicate was imputed. Correction for batch effects was performed using ComBat version: 3.38 [43]. Features missing in <20% of the samples were imputed using k-nearest neighbors and features missing in >20% were removed; we imputed less than 3% of the data points. The remaining features were log2-transformed and Z-score standardized (i.e., subtracting the metabolite-specific population mean and dividing by the metabolite-specific standard deviation). This was done to normalize the data and pool the metabolites across the HILIC-positive and C18-negative chromatography runs. We restricted our statistical investigation to the features that matched m/z within 10 ppm and retention time within 30 s of metabolites that the Emory CBL previously identified using accurate mass MS1 signal, coelution with authentic standard, and ion dissociation spectra (MS^2^/MS^n^) matching the authentic standard [44]. Meaning, we believe each analyzed metabolite met Level 1 identification according to the Metabolomics Standards Initiative [45] and Schymanski et al. [46]. This left 125 HILIC-positive and 61 C18-negative metabolites to be analyzed.

### 2.4. Statistical Analysis

Here, we summarize the steps we took to identify prognostic metabolic subtypes in the HNSCC patient population. First, we identified metabolic clusters of HNSCC patients using unsupervised hierarchical clustering of plasma metabolites, an approach that groups HNSCC patients according to metabolite patterns agnostic to patient demographics, clinical factors, or survival. Second, we determined the relative importance of each metabolite to the clustering via post-hoc Wilcoxon rank sum tests. We inputted the meaningful metabolites into pathway enrichment analysis to metabolically characterize the clusters. Third, we examined if the metabolic subtypes were associated with baseline clinical factors (e.g., age, sex, HPV status) and immune markers. Lastly, we modeled the metabolic subtypes with overall and progression-free survival to determine their potential clinical utility.

The unsupervised hierarchical clustering used 186 lab-confirmed plasma metabolites across 209 HNSCC patients. The patients were clustered using JASP version 0.14 according to Euclidean distance and Ward.D linkage, two common algorithmic settings. We fit models allowing for 2–6 patient clusters and chose the optimal model from a variety of clustering metrics: the lowest Bayesian Information Criterion, the highest silhouette scores, the highest Dunn index, the highest Calinski–Harabasz index, and the lowest model entropy. After determining the optimal model, we performed post-hoc Wilcoxon rank sum tests to rank each metabolite according to differential expression between the clusters.

We characterized each cluster via metabolic pathway enrichment analysis using roughly the top 20% of differentially expressed metabolites (i.e., 38 metabolites that met a Benjamini–Hochberg multiple testing adjusted *p*-value < 0.00001). Since we used differential expression as a post-hoc ranking tool, the top 20% cutoff is arbitrary. By limiting the pathway analysis to the top 20% we can describe enriched pathways among the most influential metabolites to the clustering. Changing this cutoff by making it higher or lower may lead to different pathway enrichment findings. Pathway enrichment was performed by Metabolanalyst using the Small Molecule Pathway Database (SMPDB) [47]. We calculated enrichment by comparing the number of pathway hits to the subset of differentially expressed metabolites relative to the expected number of pathway hits if 38 metabolites were drawn at random from the 186 total using a chi-squared test statistic.

We tested the differences in proportions or means of many clinical factors across the subtypes using chi-square or *t*-tests as appropriate. We also examined whether the metabolic subtypes were associated with patient overall survival via Kaplan–Meier curves and Cox models adjusted for age, sex, race, body mass index, marriage status, HPV status, smoking history, tumor site and stage, treatment, Eastern Cooperative Oncology Group (ECOG) performance, feeding tube, patient-reported prior comorbidities, and circulating levels of albumin, hemoglobin, neutrophil-to-lymphocyte ratio (NLR) and platelet-to-lymphocyte ratio (PLR). We tested the proportional hazards assumption by modeling a subtype-with-time interaction term. We stratified our survival analyses and Kaplan–Meier curves by whether subjects were never vs. ever smokers, then additionally by HPV status (related vs. unrelated). We classified each subject as either alive or dead at three years of follow-up, ran a logistic regression, and plotted a receiver operating characteristic (ROC) curve of three-year survival against three models: an HPV status-only model, a metabolic subtype-only model, and a model with HPV status and metabolic subtypes. Unless stated, all analyses were conducted using R version 4.2.2 [48] and RStudio build 386 [49].

### 2.5. Sensitivity Analysis

To examine the robustness of our hierarchical clustering approach, we performed two alternative clustering methods, K-means and random forest, and report those cluster groupings with overall survival.

## 3. Results

Our study population compares favorably to HNSSC patients across the U.S. as estimated by Surveillance, Epidemiology, and End Results (SEER) [50]. In our study, the median age at diagnosis was 60 years old compared to 64 in SEER. A total of 75% of our cases were men and 81% identified as white compared to 71% and 75%, respectively, in SEER. SEER estimates that 67% of HNSCC patients in the U.S. are diagnosed with regional lymph node or distant metastasis, and 68% of patients live longer than five years; our proportions were slightly higher at 74% and 77%, respectively. In our data, the average follow-up time for the subjects who died was 2.4 years compared to 5.3 years for those who were censored; only 8% of the censored subjects had fewer than 3 years of follow-up. Approximately 61% of our patients reported a current or former history of smoking, and 48% were HPV-related. Half of the tumors occurred in the oropharynx. Across many demographic and clinical characteristics, only age statistically differed between the subtypes (Table 1 and Appendix A).

A total of 186 lab-confirmed metabolites were used to cluster 209 HNSCC patients: 9 amines; 59 amino acids or amino acid-derivatives; 9 carbohydrates; 17 cholines, betaines, carnitines, or Co-As; 31 lipids; 7 nucleic acids, nucleosides, or nucleotides; 31 simple organic acid derivatives; 11 vitamins or cofactors; and 12 metabolites that do not fit those categories (Appendix A). The metabolites covered 88 different human metabolic pathways per the SPMDB reference database (Appendix A). Furthermore, the median pair-wise Pearson correlation coefficient between metabolites was 0.04 with an inter-quartile range from −0.03 to 0.12 (Appendix A), suggesting robust metabolite heterogeneity. Despite this heterogeneity, certain groups of metabolites, namely fatty acids, were highly correlated (Appendix A and Appendix A).

The optimal model, according to five of six clustering metrics (Appendix A), created two patient clusters, herein named metabolic subtypes A and B (Figure 1). We refrain from further labeling the subtypes dues to the relative, not absolute, nature of the metabolic differences. Subtypes A and B corresponded to 41% and 59% of our HNSCC study population, respectively. In Wilcoxon rank sum tests, 38 of the 186 metabolites were differentially expressed at a multiple testing adjusted *p*-value < 0.00001 (Appendix A). From those 38, the top enriched metabolic pathways were fatty acid biosynthesis (four metabolites; *p* = 0.004), transfer of acetyl groups into mitochondria (three metabolites; *p* = 0.03), arginine and proline metabolism (five metabolites; *p* = 0.06), and galactose metabolism (three metabolites; *p* = 0.07) (Appendix A). To further explain the enrichment calculation, among our 186 total metabolites, six mapped to the fatty acid biosynthesis pathway. Among those six, four were ranked in the top 38, representing a relatively rare event (χ2 = 8.15, df = 1, *p* = 0.004) if we drew 38 metabolites from the 186 at random. The metabolite Z-scores suggest relatively higher levels of biosynthesis pathways for fatty acids and amino acids, and glycolytic energy metabolism was found in samples from patients in subtype A versus subtype B (Table 2).

Relative to subtype B, subtype A patients experienced significantly worse overall and progression-free survival. The estimated 3-year overall survival was 73.3% for subtype A patients and 88.3% for subtype B; log-rank *p* = 0.003 (Figure 2a). The estimated 3-year progression-free survival was 68.2% for subtype A patients and 79.4% for subtype B; log-rank *p* = 0.05 (Appendix A). However, the survival pattern varied by smoking history. Among never smokers, overall and progression-free survival did not differ by subtype—the 3-year overall survival was 93.1% for subtype A patients and 91.3% for subtype B; log-rank *p* = 0.46 (Figure 2b and Appendix A). In contrast, among ever smokers, subtype A had significantly worse survival compared to subtype B—the 3-year overall survival was 61.0% for subtype A patients and 86.3% for subtype B; log-rank *p* = 0.001 (Figure 2c and Appendix A). Further stratifying the ever smoker population by HPV status showed that the metabolic subtype remained significantly associated with overall and progression-free survival regardless of HPV status (Figure 2d and Appendix A). Among HPV-unrelated smokers, the estimated 3-year overall survival was 50.7% for subtype A patients and 80.9% for subtype B (*p* = 0.03). Among HPV-related smokers, the estimated 3-year overall survival was 77.4% for subtype A patients and 95.8% for subtype B (*p* = 0.008). Thus, the association with survival was dependent on smoking history but indifferent to HPV status. Intriguingly, despite the importance of having a smoking history, the smoking-related metabolites nicotine (Wilcoxon rank sum *p* = 0.94) and cotinine (Wilcoxon rank sum *p* = 0.06) played almost no role in classifying a patient as subtype A or B (Appendix A).

The association of the metabolic subtypes with overall survival did not meaningfully change after adjustment by other factors (Table 3). The overall (*p* = 0.58) and progression-free (*p* = 0.37) cox models met the proportional hazards’ assumption testing a subtype by the time interaction term. In age, sex, HPV, and smoking-adjusted Cox models subtype A patients had three times the risk of mortality of subtype B (hazard ratio = 2.98, 95% CI: 1.48, 5.99) among current and former smokers. Further adjustment for race, body mass index, alcohol consumption, marital status, tumor site and stage, treatment, ECOG performance, gastronomy tube, prior comorbidities, and circulating immune markers NLR and PLR only strengthened the point estimates (HR = 3.58, 95% CI: 1.46, 8.78). Given how balanced the risk factors were between the subtypes, as shown in Table 1, it is unsurprising that covariate adjustment did not impact the association between subtype and survival. Similar to the Kaplan–Meier results, we observed no significant associations among never smokers and saw no pattern that would suggest a larger sample size would produce a significant association. The progression-free survival results followed the same pattern as the overall survival results but were slightly attenuated (Table 4). Of note, subtype A had nine times the mortality of subtype B (HR = 9.24, 95% CI: 1.64, 52.1) in age and sex-adjusted models among HPV-related smokers; however, we believe this may be better interpreted as a preliminary finding due to the small number of deaths (N = 9) in this sub-stratum.

We used logistic regression and a ROC curve to compare the ability of three models (HPV-only model vs. metabolic subtype-only model vs. HPV-plus metabolic subtype model) to predict 3-year overall survival in HNSCC patients with a history of smoking. The ROC curves in Figure 3 show that, in our data, adding pretreatment metabolic subtypes to the clinically accepted model—a model that uses only HPV—would improve the model’s classification of high-risk/low-risk by 20% (AUC of the model with metabolic subtypes = 0.73 vs. AUC of the model with HPV alone = 0.61).

In the sensitivity analyses (Appendix A), the random forest clustering showed a similar pattern to the hierarchical clustering, classifying 46% of the HNSCC population into a high-risk group and 54% into a low-risk group. Compared to the low-risk group, the high-risk group had a statistically significant risk of death among HNSCC ever smokers (fully adjusted Cox model HR = 6.75, 95% CI: 2.24, 20.3). The K-means clustering classified 42% into a high-risk group and 58% into a low-risk group. It also found a statistically significant risk of death among HNSCC current and former smokers (fully adjusted Cox model HR = 2.75, 95% CI: 1.09, 6.95).

## 4. Discussion

Using 186 lab-confirmed plasma metabolites, unsupervised hierarchical clustering parsed 209 HNSCC patients into two metabolic subtypes differentiated by the fatty acid, Co-A, amino acid, and galactose metabolism pathways. The subtypes were associated with a high-risk and low-risk of cancer progression and mortality but only among current and former smokers. Though our results should be replicated, our data suggest that a systemic metabolomic biomarker would be independent of other risk factors, most importantly HPV status, and may add crucial information to risk stratification for either HPV-related or HPV-unrelated HNSCC patients with a history of smoking. Our findings raise relevant and timely research questions about adding a non-invasive metabolomic blood biomarker as translational research in HNSCC clinical trials.

We believe this is the first study to cluster HNSCC patients by metabolites and investigate survival, complicating the task of comparing our results to the prior literature. A total of eleven HNSCC studies have investigated metabolomics in patient blood: six comparing cancer vs. controls [51,52,53,54,55,56], four comparing pre- vs. post-treatment [57,58,59,60], and one comparing extranodal extension [61], but none investigated survival. Despite the differences in approach, we see similarities between our findings and those of prior research in relation to sugar metabolism (glucose, galactose, fructose), amino acids (proline), acetyl CoA transport (citrate), and fatty acid biosynthesis (acetoacetate, β-hydroxybutyrate). Notably, Li et al.’s metabolic pathway analysis found enrichment in arginine and proline metabolism and CoA biosynthesis [56], matching our enrichment findings. In addition, Rodríguez-Tomàs et al. found advanced prognosis to be associated with pre-treatment plasma β-hydroxybutyrate [58]—an epigenetic signaling metabolite that we found crucial to our clustering.

The enriched metabolic pathways that differentiated the subtypes centered around fatty acid and amino acid biosynthesis and glycolytic metabolism. However, since individual metabolites are often linked to multiple pathways these differences likely represent broader generalities. Our strongest findings were fatty acid biosynthesis and its precursor pathway as well as the transfer of acetyl-CoA to the mitochondria—a vital pathway in the tricarboxylic acid (TCA) cycle. Fatty acid synthesis is an energy storage pathway in which carbons are added to acetyl-CoA via the enzyme fatty acid synthase (FAS) to create longer chain fatty acids necessary for cellular proliferation. In HNSCC, upregulation of FAS and its regulator, epidermal growth factor receptor (EGFR), is positively associated with HPV-negative tumor progression and metastasis [62]. Moreover, FAS and EGFR inhibitors are under active investigation as targets for HNSCC treatment and resistance [63,64]. Likewise, the arginine and proline pathway is central to the biosynthesis of key amino acids for proliferation. Li and colleagues found arginine biosynthesis and other amino acid pathways positively associated with HNSCC compared to controls [55]. Additionally, two arginine and proline pathway metabolites we observed, S-adenosylmethionine and guanidinoacetate, are the metabolic precursors to creatine and phosphocreatine, which are inversely associated with pathological extranodal extension, an adverse prognostic factor in HPV-negative HNSCC [61]. Galactose metabolism, via the Leloir pathway, converts galactose to glucose 1-phosphate to undergo glycolysis, an energy expenditure pathway that is perhaps the most studied metabolic pathway in HNSCC [65]. Wu and colleagues found galactose metabolism positively associated with HNSCC prognosis [29], and in prior research, we found it associated with HPV status and survival [66]. Uptake of glucose is also a prognostic marker in HNSCC imaging [26], and there is considerable interest in targeting or inhibiting glycolysis to improve HNSCC response to radiotherapy [67,68].

An intriguing yet unexplained finding was the prognostic dependency of the subtypes on a prior smoking history. Since the subtypes had similar proportions of current, former, and never smokers, this means their association with survival was dependent on but not confounded by smoking. Unfortunately, the questionnaires implemented by the parent study did not collect data to investigate interaction across pack-years but we note that levels of the smoking-related metabolites nicotine and cotinine did not play a role in subtype clustering and were not associated with survival. We do not yet have a biological rationale for this finding. However, in a somewhat parallel finding, Gutkind et al.’s phase II trial reported a successful histological response to the metabolic diabetes drug metformin only among current and former smokers [69]. Gutkind and colleagues showed that the response was linked to suppression of the mammalian target of the rapamycin (mTOR) pathway, a proliferative pathway that regulates fatty acid biosynthesis [70] and acetyl-CoA [71] but is also regulated by arginine metabolism [72]. There are ongoing and recently completed phase II clinical trials investigating the safety and efficacy of adding metformin to improve HNSCC patient response to standard chemoradiotherapy or immunotherapy [73,74]. For those trials, and future HNSCC clinical trials of metformin or other mTOR inhibitors, we believe a metabolomic biomarker would be a valuable translational addition.

The prognostic ability of the metabolic subtypes was independent of HPV status, and this may be particularly pertinent to HPV-related HNSCC patients with an extensive smoking history. HPV-related HNSCC smokers are an intermediate-risk group whose prognosis appears too uncertain to test de-escalated treatment [10,75]. Among this group of patients, metabolic subtype A showed a nearly ten-fold change in survival relative to subtype B. Though our analysis does not disentangle why smoking contributes to worse outcomes in HPV-related disease, it does suggest that a metabolomic biomarker might aid clinical risk stratification for this varied patient group.

A specific limitation of the study was the inability to assess disease-induced cachexia, muscle loss, or another abnormal nutritional state that may cause metabolic dysregulation. We did not have information on muscle mass, nor did we have information on weight loss prior to study entry; however, the average body mass index between the metabolic subtypes was similar at baseline and that similarity did not change during the first year of follow-up. Additionally, patients were not instructed to fast before blood collection, and we had limited hematological variables from which to assess anemia or nutrition deficiency; we did not have serum hematocrit, bilirubin, iron, ferritin, lactate dehydrogenase, or aminotransaminases. The estimated prevalence of pretreatment cachexia in HNSCC is upwards of 20% and it is associated with poor prognosis and mortality [76], making it a potential confounder. We attempted to mitigate this confounding by adjusting the survival models by factors correlated with a patient’s nutritional state, such as body mass index, albumin and hemoglobin levels, as well as the presence of a gastronomy tube but we cannot rule out residual confounding. Metabolomics may be intimately involved with cancer patient cachexia. In a very small number of HNSCC patients, Boguszewicz et al. showed evidence that increased serum levels of ketone bodies (β-hydroxybutyrate, acetone, and acetoacetate) along with decreased serum glucose may provide a real-time signal of treatment-related toxicity and future cachexia [77]. Though we used pretreatment LC-MS metabolomics and Boguszewicz et al. used posttreatment ^1^H NMR metabolomics, the similarities between our two findings warrant further investigation.

Other limitations include a relatively small sample size, the lack of external validation, and the inability to link our plasma results with a tumor. Though a sample size of 200 is not particularly small, the stability and reproducibility of unsupervised clustering are improved with larger numbers and multiple datasets. Unfortunately, unlike genomics datasets, metabolomics datasets are scarce and not publicly available. Our results need to be validated, and we are actively working to do so. However, since we were analyzing relative intensity rather than metabolite concentration further standardization would be required before pooling data with another study. The recruitment protocol of the parent study limited our metabolomics analyses to patient plasma. Without paired tumor specimens, we cannot investigate tumor characteristics such as p53 or Ki67 staining. We also cannot directly compare our circulating metabolic subtypes to tumor genomic subtypes found in the prior literature. Moreover, we cannot be sure circulating metabolites adequately represent the metabolism occurring within the tumor and its microenvironment. Therefore, our study is more akin to biomarker discovery from systemic metabolic signals rather than the metabolic classification of HNSCC. However, it is noteworthy that the metabolic pathways we found in patient plasma were also found in tumor genomic subtypes in prior research [78]. Limiting our analysis to lab-confirmed metabolites mitigated issues with metabolite annotation but opens the possibility of missing important metabolic signals. Still, with 186 confirmed metabolites clustering 209 patients, there is a potential issue of high dimensional data making the clusters less reproducible. We explored reproducibility using alternative clustering algorithms, random forest and k means clustering, and we found similar results in both to our primary findings.

## 5. Conclusions

In a first-of-its-kind investigation, we believe our study provides compelling results to consider metabolomics as a translational prognostic biomarker of HPV-related and HPV-unrelated HNSCC tobacco users. However, there are still pressing research gaps to fill regarding independent validation, integration with other omics data, metabolic interventions, and incorporation into clinical trials before the potential for metabolomics is fully recognized. Addressing these gaps is crucial to gain a fuller understanding of how metabolomics may aid in the early detection, prognosis, treatment monitoring, and targeted therapies of HNSCC.

## Figures and Tables

**Figure 1 cancers-15-03184-f001:**
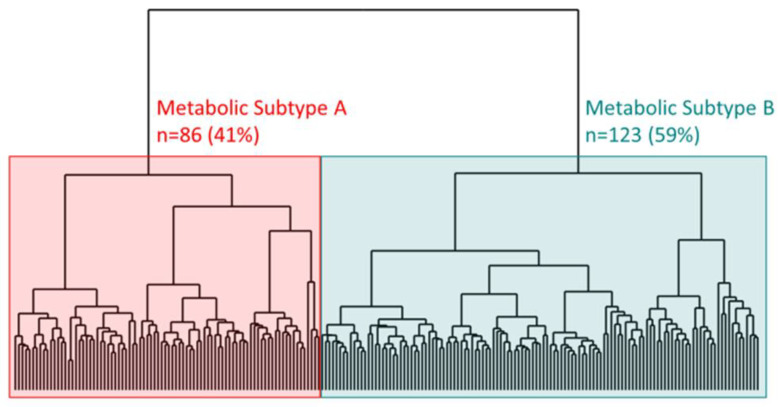
Dendrogram showing two metabolic subtypes, A (red) and B (teal), identified from an unsupervised hierarchical clustering analysis of 209 head and neck cancer patients using 186 lab-confirmed plasma metabolites. Subtype A, the high-risk type showing relatively higher amounts of fatty acid, acetyl Co-A, and amino acid biosynthesis as well as glycolytic metabolism, represents 41% of the sample population and subtype B, the remaining 59%.

**Figure 2 cancers-15-03184-f002:**
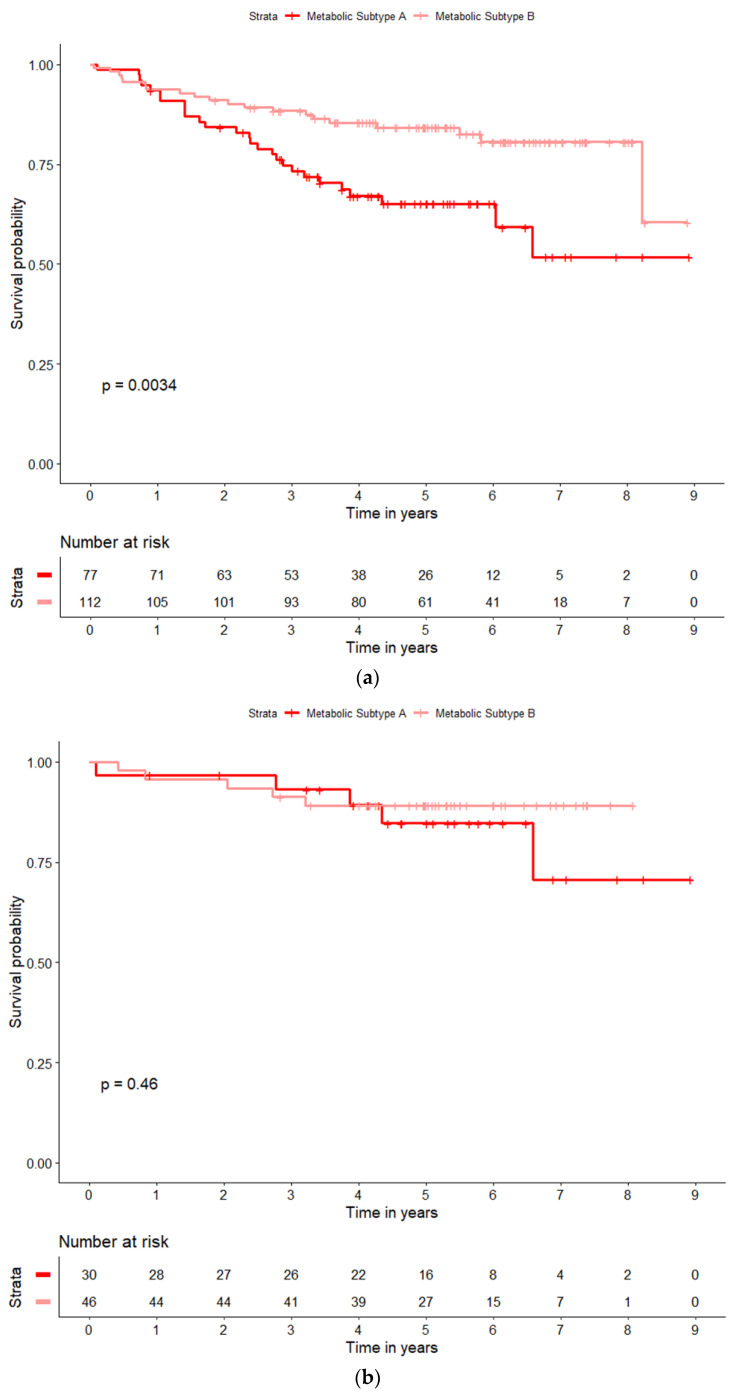
(**a**–**d**). Kaplan–Meier overall survival curves by Metabolic Subtypes A and B. (**a**) is among the full population (N = 47 deaths) in which the estimated 3-year survival is 73.3% for Subtype A (dark red) and 88.3% for Subtype B (light red); the curves are statistically significant with a log-rank *p*-value = 0.003. (**b**) shows the metabolic subtype overall survival curves (A vs. B) amongst never smokers (N = 10 deaths). The estimated 3-year survival is 93.1% for subtype A (dark red), and 91.3% for subtype B (light red); *p* = 0.46. (**c**) shows the metabolic subtype overall survival curves (A vs. B) amongst ever smokers (N = 37 deaths). The estimated 3-year survival is 61.0% for subtype A (dark red), and 86.3% for subtype B (light red); *p* = 0.001. (**d**) further stratifies the survival curves by HPV status (unrelated and related) amongst ever smokers (N = 37 deaths). The estimated 3-year survival for the four groups is 50.7% for Subtype A, HPV-unrelated (dark red); 80.9% for Subtype B, HPV-unrelated (light red); 77.4% for subtype A, HPV-related (dark blue); and 95.8% for subtype B, HPV-related (light blue).

**Figure 3 cancers-15-03184-f003:**
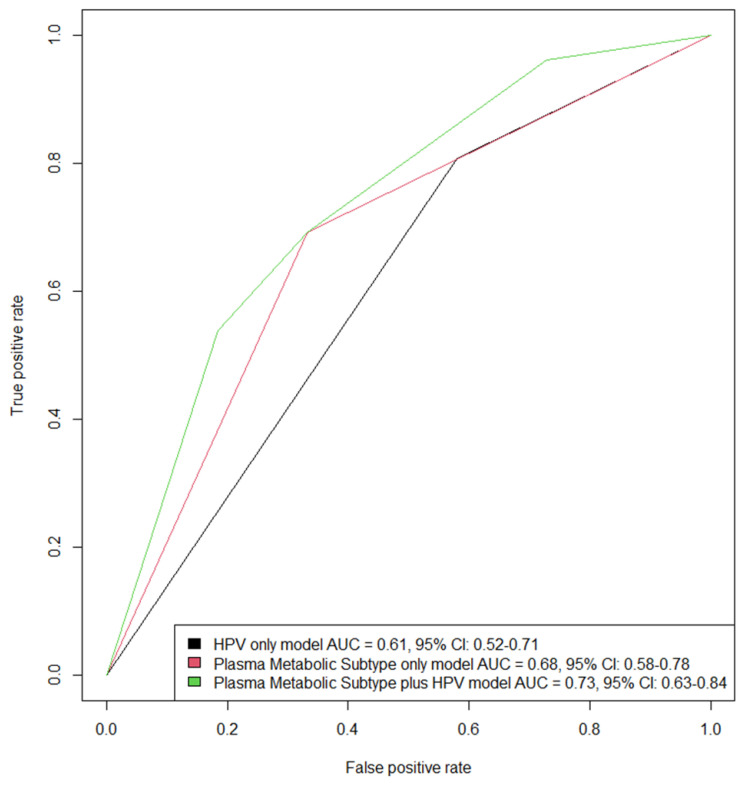
Receiver Operating Characteristic (ROC) curves showing the ability of three logistic regression models to classify death within three years of follow-up for head and neck cancer patients with a prior smoking history. The first model (black line) uses only HPV status. The second model (red line) uses only the pretreatment plasma metabolomic subtype biomarker. The third model (green line) adds the metabolomic subtype to a model of HPV status. Comparing the area under the curve (AUC) of the first and third models indicates a 20% relative increase in predictive ability when the metabolomic subtype biomarker is added to a model of HPV status (0.73/0.61 = 1.197).

**Table 1 cancers-15-03184-t001:** Descriptive table of 209 HNSCC patients stratified by metabolic subtypes A and B derived from hierarchical clustering of 186 plasma metabolites.

		All (N = 209)	Subtype A (N = 86)	Subtype B (N = 123)	*p* Value
Age	mean (SD)	59.3 (10.1)	61.7 (10.3)	57.6 (9.7)	0.004
BMI	mean (SD)	27.5 (5.2)	28.1 (5.8)	27.1 (4.7)	0.18
Sex	Male	156 (75%)	65 (76%)	91 (74%)	0.92
	Female	53 (25%)	21 (24%)	32 (26%)	
Race	White	170 (81%)	73 (85%)	97 (79%)	0.36
	Black	39 (19%)	13 (15%)	26 (21%)	
HPV status	Negative	108 (52%)	43 (50%)	65 (53%)	0.79
	Positive	101 (48%)	43 (50%)	58 (47%)	
Smoking history	Never	81 (39%)	33 (39%)	48 (39%)	0.83
	Former	67 (33%)	29 (35%)	38 (31%)	
	Current	58 (28%)	22 (26%)	36 (30%)	
Alcohol	<1 drink/week	114 (55%)	47 (55%)	67 (55%)	0.99
	1+ drink/week	92 (45%)	38 (45%)	54 (45%)	
Marital status	Married or partnered	148 (71%)	62 (72%)	86 (70%)	0.85
	Single	61 (29%)	24 (28%)	37 (30%)	
Tumor site	Oropharynx	106 (51%)	45 (53%)	61 (50%)	0.56
	Oral cavity	31 (15%)	12 (14%)	19 (15%)	
	Larynx	35 (17%)	11 (13%)	24 (20%)	
	Other	36 (17%)	17 (20%)	19 (15%)	
Tumor stage *	I	10 (5%)	5 (6%)	5 (4%)	0.77
	II	16 (8%)	5 (6%)	11 (9%)	
	III	83 (40%)	36 (42%)	47 (39%)	
	IV	98 (47%)	40 (47%)	58 (48%)	
T	1	37 (18%)	19 (22%)	18 (15%)	0.43
	2	51 (25%)	20 (24%)	31 (26%)	
	3	47 (23%)	21 (25%)	26 (22%)	
	4	70 (34%)	25 (29%)	45 (37%)	
N	0	46 (22%)	18 (21%)	28 (23%)	0.09
	1	22 (11%)	4 (5%)	18 (15%)	
	2	130 (63%)	60 (70%)	70 (58%)	
	3	8 (4%)	4 (5%)	4 (3%)	
Treatment	Radiotherapy	45 (22%)	20 (23%)	25 (20%)	0.87
	Chemoradiotherapy with Cisplatin	118 (56%)	48 (56%)	70 (57%)	
	Chemoradiotherapy with Carboplatin and Paclitaxel	46 (22%)	18 (21%)	28 (23%)	
Feeding tube	No	80 (40%)	35 (43%)	45 (39%)	0.60
	Yes	118 (60%)	46 (57%)	72 (62%)	
ECOG performance	Active	100 (50%)	39 (48%)	61 (52%)	0.89
	Restricted	73 (37%)	31 (38%)	42 (36%)	
	Non-working	26 (13%)	11 (14%)	15 (13%)	
Prior comorbidity	Yes	154 (75%)	66 (77%)	88 (73%)	0.69
	No	52 (25%)	20 (23%)	32 (27%)	
Albumin	mean (SD)	3.95 (0.42)	3.97 (0.39)	3.94 (0.43)	0.60
Hemoglobin	mean (SD)	13.18 (1.80)	13.22 (1.71)	13.15 (1.87)	0.80
Neutrophil-to-lymphocyte ratio	mean (SD)	3.18 (2.07)	3.32 (1.98)	3.08 (2.13)	0.45
Platelet-to-lymphocyte ratio	mean (SD)	170,347 (98,859)	161,891 (64,755)	176,207 (116,774)	0.35

HNSCC = head and neck squamous cell carcinoma, BMI = body mass index, HPV = human papillomavirus, ECOG = Eastern Cooperative Oncology Group, SD = standard deviation. * Using the 8th edition of the American Joint Committee on Cancer Tumor Staging by Site.

**Table 2 cancers-15-03184-t002:** Fourteen lab-confirmed metabolites that comprised the enriched metabolic pathways that differentiated subtypes A and B.

Name	mz	rt	ESI	Adduct	HMDB#	Z-Score * A	Z-Score * B	*p*-Value **
Fatty acid biosynthesis (*p* = 0.004)								
Acetoacetate	101.0244	25	C18-	(M − H)	HMDB0000060	0.53	−0.37	2.0 × 10^−9^
β-hydroxybutyrate	103.0401	22	C18-	(M − H)	HMDB0000357	0.43	−0.30	4.3 × 10^−6^
FA 16:0 (Palmitate)	255.2329	231	C18-	(M − H)	HMDB00220	0.38	−0.27	4.6 × 10^−6^
FA 14:0 (Myristate)	227.2016	212	C18-	(M − H)	HMDB00806	0.38	−0.27	7.1 × 10^−6^
Transfer of acetyl groups into the mitochondria (*p* = 0.03)								
Glucose	215.0328	21	C18-	(M + Cl)	HMDB0000122	0.40	−0.28	3.1 × 10^−6^
Citric acid	191.0197	19	C18-	(M − H)	HMDB0000094	0.41	−0.28	7.5 × 10^−6^
Malic acid	133.0143	20	C18-	(M − H)	HMDB0000156	0.37	−0.26	7.6 × 10^−6^
Arginine and Proline metabolism (*p* = 0.06)								
S-adenosylmethionine	399.1445	162	HILIC+	(M + H)	HMDB0001185	0.47	−0.33	1.6 × 10^−9^
Proline	116.0706	87	HILIC+	(M + H)	HMDB0000162	0.49	−0.35	8.3 × 10^−8^
Ornithine	133.0972	125	HILIC+	(M + H)	HMDB0000214	0.45	−0.32	5.1 × 10^−7^
Citrulline	176.103	109	HILIC+	(M + H)	HMDB0000904	0.40	−0.28	8.4 × 10^−7^
Guanidinoacetate	118.0617	89	HILIC+	(M + H)	HMDB0000128	0.41	−0.29	8.8 × 10^−7^
Galactose metabolism (*p* = 0.07)								
Fructose	219.0265	73	HILIC+	(M + K)	HMDB0000660	0.60	−0.42	3.6 × 10^−11^
Galactose	203.0526	71	HILIC+	(M + Na)	HMDB0000143	0.59	−0.41	3.6 × 10^−11^
Glucose	215.0328	21	C18-	(M + Cl)	HMDB0000122	0.40	−0.28	3.1 × 10^−6^

The mz = mass-to-charge ratio, rt = retention time, ESI = electrospray ionization method, FA = fatty acid, HMDB = human metabolome database. * Z-scores were calculated by subtracting the metabolite-specific population mean and dividing by the standard deviation. ** Wilcoxon rank sum *p*-values adjusted for multiple testing via the Benjamini–Hochberg method.

**Table 3 cancers-15-03184-t003:** Estimated associations of metabolic subtypes A vs. B (referent) with patient overall survival via Cox models among the full HNSCC population and stratified by smoking status.

	HR	95% CI	*p* Value
Full Population (N = 189, N_deaths_ = 47)			
Unadjusted model	2.33	(1.30, 4.16)	0.004
Age, sex, HPV, smoking adjusted model	2.38	(1.30, 4.33)	0.005
Fully adjusted model ^a^	2.76	(1.32, 5.77)	0.007
Among Ever Smokers (N = 113, N_deaths_ = 37)			
Unadjusted model	2.91	(1.47, 5.78)	0.002
Age, sex, HPV adjusted model	2.98	(1.48, 5.99)	0.002
Fully adjusted model ^a^	3.58	(1.46, 8.78)	0.005
Among Never Smokers (N = 76, N_deaths_ = 10)			
Unadjusted model	1.60	(0.46, 5.51)	0.46
Age, sex, HPV adjusted model	1.13	(0.30, 4.24)	0.85
Fully adjusted model ^a^	0.92	(0.06, 15.3)	0.95

^a^ Further adjusted for race, body mass index, alcohol history, marital status, tumor site and stage, treatment, Eastern Cooperative Oncology Group (ECOG) performance, gastronomy tube, prior comorbidities, albumin, hemoglobin, neutrophil-to-lymphocyte ratio, and platelet-to-lymphocyte ratio; 40 subjects (25 ever, and 15 never smokers), including 5 deaths (4 ever, and 1 never smokers), were removed due to missing covariates. HNSCC = head and neck cancer squamous cell carcinoma, HR = hazard ratio, CI = confidence interval, HPV = human papillomavirus.

**Table 4 cancers-15-03184-t004:** Estimated associations of metabolic subtypes A vs. B (referent) with progression-free survival via Cox models among the full HNSCC population and stratified by smoking status.

	HR	95% CI	*p* Value
Full Population (N_events_ = 62)			
Unadjusted model	1.62	(0.98, 2.67)	0.06
Age, sex, HPV, smoking adjusted model	1.65	(0.99, 2.75)	0.06
Fully adjusted model ^a^	1.70	(0.93, 3.11)	0.08
Among Ever Smokers (N_events_ = 47)			
Unadjusted model	2.06	(1.14, 3.71)	0.02
Age, sex, HPV adjusted model	2.13	(1.17, 3.89)	0.01
Fully adjusted model ^a^	2.11	(1.03, 4.32)	0.04
Among Never Smokers (N_events_ = 15)			
Unadjusted model	1.00	(0.35, 2.80)	0.99
Age, sex, HPV adjusted model	0.80	(0.27, 2.37)	0.68
Fully adjusted model ^a^	0.48	(0.08, 3.03)	0.44

^a^ Further adjusted for race, body mass index, alcohol history, marital status, tumor site and stage, treatment, Eastern Cooperative Oncology Group (ECOG) performance, gastronomy tube, prior comorbidities, albumin, hemoglobin, neutrophil-to-lymphocyte ratio, and platelet-to-lymphocyte ratio; 40 subjects (25 ever and 15 never smokers), including 5 events (4 ever and 2 never smokers), were removed due to missing covariates. HNSCC = head and neck cancer squamous cell carcinoma, HR = hazard ratio, CI = confidence interval, HPV = human papillomavirus.

## Data Availability

The data are available on request to the corresponding author. De-identified metabolomics data will be deposited at the UC San Diego Metabolomics Workbench. Computer code for the metabolomics analysis using xmsPANDA is freely available at https://github.com/kuppal2/ (accessed on 28 February 2023). Code for the clustering and survival analyses are available upon request to the corresponding author.

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
