# Peer review of "Unsupervised Hierarchical Clustering of Head and Neck Cancer Patients by Pre-Treatment Plasma Metabolomics Creates Prognostic Metabolic Subtypes"

_cancers, 2023, doi:10.3390/cancers15123184_

Round 1

Reviewer 1 Report

The analytical result (clustering with metabolomics data) looks like having prognostic prediction ability. However, the data analyses, results interpretation, and conclusion look optimistic.

The number of patients was too small for such analyses.

Therefore, the authors should validate the analytical results with independent datasets. Otherwise, the data should be split into two data. One of the data should be used for similar analyses. The latter should be used as validation data.

The quality of the metabolomics data was not adequately presented.  

Line 166. The authors claimed the level 1 identification level of MSI. However,at least the author should conduct a spiking test to confirm the identification.

Line 159. “Features missing in <20% of the samples were imputed using k-nearest neighbors; features missing in >20% were removed.”

This imputation can be used depending on the reason for the missing values. The use of k-NN depends on the imputation using other sample data. If all measurement and data processing are correctly conducted, the missing indicate the not/detected, i.e., the concentration=0.

Line 158. “Correction for batch effects was performed using ComBat.[43]”

The data may be measured within multiple batches. The data without correction also should be presented.

The linearity between peak intensities and metabolite concentrations with lower and upper limits should be presented. The data used for subsequent analyses should range within these limits. The samples should be diluted with various dilution ratios, and the linearity must be evaluated.

Because subsequent analyses, such as clustering, strongly depend on the linearity quality. The triplicate measurement is a different topic.

L136-146 The section “High-resolution untargeted metabolomics (HRM) of blood plasma:” is not scientific. There was no preprocessing protocol. If internal standards were not injected when the metabolites were extracted, the raw signal might fluctuate depending on the sensitivity of the mass detector. When the internal standard are used, raw data should be normalized by these internal standards. However, no descriptions.

L147-168. Most of the data processing described here is not reproduceible. Each process has various options. For example, the apLCMS, ComBAT has many options. How do readers reproduce this analysis?

Line 160. “The remaining features were log2-transformed and Z-score standardized”

This standardization eliminates the size information of each peak. Usually, a relatively large peak (lower than the upper linearity limit) is more reproducible. The peaks close to the lower limit are less reproducible. However, there is no linearity evaluation in this study. Therefore, we do not understand whether this standardization is suitable or not.

The age shows a significant difference between the two clusters. The effect of age should be evaluated.

In conclusion, there are severe problems in metabolomics analyses. The authors lacks the evaluation of the validity of the data, especially the reproducibility and linearity of the data. Most of the data processing was presented without any policy. There were many data corrections for missing imputation and batch-effect corrections. A lot of problems exist in the data.

Author Response

REVIEWER #1 Comments and Suggestions for Authors

We appreciate the reviewer’s thoughtful comments and suggestions. We address in bold each point individually and have made relevant changes to the manuscript where we can.

The analytical result (clustering with metabolomics data) looks like having prognostic prediction ability. However, the data analyses, results interpretation, and conclusion look optimistic.

The number of patients was too small for such analyses.

Therefore, the authors should validate the analytical results with independent datasets. Otherwise, the data should be split into two data. One of the data should be used for similar analyses. The latter should be used as validation data.

We appreciate the reviewer’s suggestion to validate our findings in an independent dataset. Unfortunately, we are unaware of any publicly available HNSCC metabolomics datasets large enough with survival follow-up to do so; I believe our data and this analysis is the first. We are, however, in the process of obtaining an external cohort to validate our findings but that will take time. We now note a smaller sample size as a primary limitation of our study, but again, we are unaware of any dataset that is larger.

We would like to point out a couple of issues about the reviewer’s suggestions to split the data. First, it would create even smaller datasets than the one we have, magnifying any prior issues with sample size. Second, while splitting a data into a training and validation sets is paramount to a supervised learning approach, it is less clear how much it helps in an unsupervised approach, such as the one we have done. Independent replication with an external cohort would serve a better method of validation for an unsupervised approach, and as mentioned, we are in the process of obtaining an external cohort. Moreover, since this was an unsupervised clustering approach, the survival data (death and follow-up time) played no role in the construction of the metabolomic subtypes, thus the mortality findings do not need internal validation but rather external replication, like all research. Finally, we do recognize the uncertainty surrounding an unsupervised hierarchical clustering and that is why we conducted sensitivity analyses with two other clustering approaches, random forest and Kmeans. These approaches use very different clustering algorithms and in both we found similar results to our hierarchical clustering approach: the population was divided into two similar proportions of high-risk/low-risk groups that were significantly associated with survival among current and former smokers. We have further clarified these sensitivity analyses in the results and discussion sections. 

The quality of the metabolomics data was not adequately presented.  

Line 166. The authors claimed the level 1 identification level of MSI. However,at least the author should conduct a spiking test to confirm the identification.

In lines 166-167 we now clarify how the lab previously identified each metabolites to MSI Level 1 that includes identified using accurate mass MS1 signal, coelution with authentic standard, and ion dissociation spectra (MS2/MSn) matching authentic standard.

Line 159. “Features missing in <20% of the samples were imputed using k-nearest neighbors; features missing in >20% were removed.”

This imputation can be used depending on the reason for the missing values. The use of k-NN depends on the imputation using other sample data. If all measurement and data processing are correctly conducted, the missing indicate the not/detected, i.e., the concentration=0.

We acknowledge that while it is possible for the intensity of a feature to be zero, we believe that a more likely event is that the value is not zero but lower than the limit of detection. In that scenario, imputation is a valid approach. With uncertainty regarding a missing data point, K nearest neighbors is a common imputation method used in metabolomics and other omics fields. We also argue that keeping zero values would prevent any transformation and unnecessarily skew any standardization, both of which are common analytical techniques in untargeted metabolomics. However, in response to this comment, we have now added a column in Supplementary Table 1 that shows the number of data points imputed for each metabolite. Overall, we imputed 2.8% of the total data points and 1.2% of the data points among the 38 metabolites that comprised our pathway analysis results.

Line 158. “Correction for batch effects was performed using ComBat.[43]”

The data may be measured within multiple batches. The data without correction also should be presented.

We fail to understand why it would be beneficial to the reader to conduct the analysis and present findings without correcting for batch effects. If the findings were the same or highly similar one could conclude that batch effects had little effect on the outcome. If the findings were materially different, then the logical conclusion would be that batch effects played a role and would need to be corrected for as we did.

The linearity between peak intensities and metabolite concentrations with lower and upper limits should be presented. The data used for subsequent analyses should range within these limits. The samples should be diluted with various dilution ratios, and the linearity must be evaluated.

Because subsequent analyses, such as clustering, strongly depend on the linearity quality. The triplicate measurement is a different topic.

We did not measure metabolite concentration. This was an untargeted analysis that extracted relative ion intensity for many features. We then restricted the analysis to the features that closely matched mz and retention time of previously confirmed metabolites by the lab. We were not analyzing the raw peak data, but rather the intensity feature table. We now include the detailed lab protocols used to extract the feature intensities from the raw data.

L136-146 The section “High-resolution untargeted metabolomics (HRM) of blood plasma:” is not scientific. There was no preprocessing protocol. If internal standards were not injected when the metabolites were extracted, the raw signal might fluctuate depending on the sensitivity of the mass detector. When the internal standard are used, raw data should be normalized by these internal standards. However, no descriptions.

L147-168. Most of the data processing described here is not reproduceible. Each process has various options. For example, the apLCMS, ComBAT has many options. How do readers reproduce this analysis?

We have now included the Emory Clinical Biomarkers Laboratory standard operating procedure for this project as a supplementary file. It includes all the inputs and code for the above programs.

Line 160. “The remaining features were log2-transformed and Z-score standardized”

This standardization eliminates the size information of each peak. Usually, a relatively large peak (lower than the upper linearity limit) is more reproducible. The peaks close to the lower limit are less reproducible. However, there is no linearity evaluation in this study. Therefore, we do not understand whether this standardization is suitable or not.

The log2-transformation and Z-score standardization was done for two primary reasons. First, it was an attempt to normalize the data as metabolomics data can often be skewed. Second, particularly the Z-score standardization was done to pool the HILIC and C18 chromatography data together. Since we extracted relative ion intensity rather than metabolite concentration, we believe the transformation and standardization are common analytical approaches performed before statistical analysis. We now specify the rationale in the methods section.

The age shows a significant difference between the two clusters. The effect of age should be evaluated.

Age was the only variable to show significant differences across the clusters, but it was only mildly associated with survival in the age, sex, HPV and smoking Cox survival models (P=0.07). In supplementary figure 1, we provide box plots of age stratified by the subtype to better show the distributions. In Table 3, we can see that age, along with sex, HPV and smoking, are weak confounders because they change the association between the metabolic subtype and mortality by a minimal amount (2.1%). Originally, we did not consider age as a possible interacting variable but explored it now and found no significant interaction (P=0.83).

In conclusion, there are severe problems in metabolomics analyses. The authors lacks the evaluation of the validity of the data, especially the reproducibility and linearity of the data. Most of the data processing was presented without any policy. There were many data corrections for missing imputation and batch-effect corrections. A lot of problems exist in the data.

Reviewer 2 Report

Minor comments:

1. Since the metabolomics was performed on patients' blood samples, more patients' data should be added:

Tumor grade? Ki67 level?

Hemoglobin level

Hematocrit level

MCV. MCH, MCHC levels

Reticulocyte count

Total and direct bilirubin levels

Serum iron

Serum LDH

Serum ferritin

AST and ALT levels

Author Response

REVIEWER #2 Comments and Suggestions for Authors

  1. Since the metabolomics was performed on patients' blood samples, more patients' data should be added:

We thank the reviewer for the thoughtful comments and suggested list of additional variables to consider. We address each point individually in bold after a general statement here. Unfortunately, since this is a secondary analysis, we do not have data on many of the suggested variables. Those that we have, we have now included into the analysis. For those variables that we do not have, we looked for prior evidence linking those variables with head and neck cancer prognosis or survival. If we found said evidence, we now include that variable as a limitation.

Tumor grade? We do not have tumor grade because the new TMN staging of HNSCC is more clinically important and has a stronger association with survival. Tumor stage also tends to be moderately to highly correlated with tumor grade. Investigating tumor grade with HNSCC survival we believe it has mixed evidence at best.

Ki67 level? Since we do not have tumor specimen, we do not have Ki67 staining data. We did find a prior association with prognosis and now include this as a limitation.

Hemoglobin level. We do have hemoglobin (gm/dL) and now include as a covariate along with albumin. Neither hemoglobin nor albumin changed our point estimate much.

Hematocrit level. We do not have hematocrit level, but we do have hemoglobin, and albumin levels and have now included them in the analysis. We now note this as a hematological limitation.

MCV. MCH, MCHC levels. We don’t have this level of detail with hemoglobin. We note this as a general limitation only have hemoglobin as a hematological variable.

Reticulocyte count. We do not have reticulocyte count. We found no association between reticulocyte count and HNSCC prognosis.

Total and direct bilirubin levels. We do not have total or direct bilirubin. We now note this as a hematological limitation.

Serum iron. We do not have iron levels. We now note this as a hematological limitation.

Serum LDH. We do not have LDH. We now note this as a hematological limitation.

Serum ferritin. We do not have ferritin. We now note this as a hematological limitation.

AST and ALT levels. We do not have AST or ALT. We now note this as a hematological limitation.

Reviewer 3 Report

This paper has many excellent qualities addresses a very important clinical need.  The study involved a unique and significantly powered prospective patient cohort. The statistical approaches are sound, but the implementation is not clear and further (re)analyses are required.  The metabolomics data was not carefully analyzed and the will require a great deal more attention.  Attached are the specific details of my review.    

Author Response

We address the review comments in the attached word document.

Round 2

Reviewer 2 Report

No more comments

Author Response

We thank the reviewer for their second review of our manuscript. Since there were no additional comments, we do not have additional changes.